# Transcriptional regulation of genes bearing intronic heterochromatin in the rice genome

**Nino A. Espinas**[1,2], **Le Ngoc Tu**[1], **Leonardo Furci**[1], **Yasuka Shimajiri**[1,3], **Yoshiko Harukawa**[1], **Saori Miura**[1], **Shohei Takuno**[4], **Hidetoshi Saze**[1]*

1 Plant Epigenetics Unit, Okinawa Institute of Science and Technology Graduate University, Onna-son, Okinawa, Japan, 2 Plant Immunity Research Group, RIKEN Center for Sustainable Resource Science (CSRS), Yokohama city, Kanagawa, Japan, 3 EditForce, Fukuoka, Japan, 4 Department of Evolutionary Studies of Biosystems, SOKENDAI (The Graduate University for Advanced Studies), Hayama, Kanagawa, Japan

* hidetoshi.saze@oist.jp

**Data Availability Statement:** All the sequence data reported in this study have been deposited in the DDBJ Sequence Read Archive under accession ID

## Abstract

Intronic regions of eukaryotic genomes accumulate many Transposable Elements (TEs). Intronic TEs often trigger the formation of transcriptionally repressive heterochromatin, even within transcription-permissive chromatin environments. Although TE-bearing introns are widely observed in eukaryotic genomes, their epigenetic states, impacts on gene regulation and function, and their contributions to genetic diversity and evolution, remain poorly understood. In this study, we investigated the genome-wide distribution of intronic TEs and their epigenetic states in the *Oryza sativa* genome, where TEs comprise 35% of the genome. We found that over 10% of rice genes contain intronic heterochromatin, most of which are associated with TEs and repetitive sequences. These heterochromatic introns are longer and highly enriched in promoter-proximal positions. On the other hand, introns also accumulate hypomethylated short TEs. Genes with heterochromatic introns are implicated in various biological functions. Transcription of genes bearing intronic heterochromatin is regulated by an epigenetic mechanism involving the conserved factor OsIBM2, mutation of which results in severe developmental and reproductive defects. Furthermore, we found that heterochromatic introns evolve rapidly compared to non-heterochromatic introns. Our study demonstrates that heterochromatin is a common epigenetic feature associated with actively transcribed genes in the rice genome.

## Author summary

Intronic regions of eukaryotic genomes accumulate many Transposable Elements (TEs) and repeats. These intronic repeats are often targeted by epigenetic silencing mechanisms and form a repressive heterochromatin structure, even within transcriptionally active genes. However, the distribution of TEs in the intragenic regions, and their contributions to genetic diversity and evolution in plant genomes, remain poorly understood. In this study, we investigated the genome-wide distribution of intronic TEs and their epigenetic states in the *Oryza sativa* genome, where TEs comprise 35% of the genome. We found

DRA008322. All other data are within the manuscript and its Supporting Information files.

**Funding:** This work was supported by MEXT Grant-in-Aid for Scientific Research on Innovative Area (http://www.mext.go.jp/a_menu/shinkou/hojyo/1218181.htm) Grant Number 19H05272 to HS, and also supported by Okinawa Institute of Science and Technology Graduate University (https://www.oist.jp) to HS. The funders had no role in study design, data collection and analysis, decision to publish, or preparation of the manuscript.

**Competing interests:** The authors have declared that no competing interests exist.

that over 10% of rice genes contain introns associated with repressive heterochromatin. Genes with heterochromatic introns are implicated in various biological functions. The conserved protein OsIBM2 is required for proper transcription of a group of heterochromatin-containing genes. We also found that heterochromatic introns evolve rapidly compared to non-heterochromatic introns. Our study indicates that heterochromatin is a common feature in transcribed genes in the rice genome.

## Introduction

Genomes of eukaryotes contain substantial numbers of transposable elements (TEs), which shape genomic structures and epigenomic landscapes [1, 2]. In plants, genomic TE contents are strongly correlated with genome size expansion [3]. Since TE insertions in genes disrupt coding sequences and regulatory elements, TEs are evolutionarily purged from genic regions and accumulated in the gene-poor pericentromeric regions of chromosomes, especially in species with small genomes, such as *Arabidopsis thaliana* [2, 4]. However, in plants with larger genomes, TEs are also distributed across the gene-rich chromosome arm regions, and often affect transcription of surrounding genes [2, 5–8].

Due to their harmful effects in the genome, TEs are often epigenetically modified and transcriptionally silenced by genome defense mechanisms [9, 10]. In plants, interdependent chromatin modifications including DNA methylation, histone modifications, and RNA interference (RNAi) play key roles in transcriptional repression of TEs. DNA cytosine methylation is found in both CG and non-CG (CHG, CHH; H = A, T, C) contexts in plant genomes, which is important in TE silencing. CG methylation is maintained through DNA replications by the Methyltransferase 1 (MET1) [11–14]. In addition, DNA methylation is directed by RNAi-based RNA-dependent DNA methylation, where small interfering RNAs (siRNAs) recruit *de novo* DNA methyltransferase to target sequences [15]. Furthermore, histone modifications, including histone H3 Lys9 methylation (H3K9me), are tightly linked to non-CG methylation, and are associated with repressive chromatin states [16]. Chromatin with these modifications results in the formation of a condensed repressive chromatin structure called heterochromatin [16–18], commonly associated with most TE sequences. The chromatin remodeler Decrease in DNA Methylation 1 (DDM1) is required for the maintenance of heterochromatin [19–21].

The formation of heterochromatin on TEs in genic regions causes transcriptional repression of surrounding genes in plant genomes [22, 23]. For example, heterochromatin associated with TEs and repetitive sequences in promoter regions often causes transcriptional repression of downstream genes [24–27]. Many TEs are also present in intronic regions especially in large plant genomes [28–31], likely due to less adverse effects on coding sequences compared to exonic insertions. Enigmatically, intronic TE sequences can also be targeted by repressive chromatin modifications, thus forming heterochromatic structure within transcription permissive chromatin environments [32]. In *Arabidopsis thaliana*, nuclear proteins, including INCREASE IN BONSAI METHYLATION 2 (IBM2)/ANTI-SILENCING1/SHOOT GROWTH1, are required for proper transcription of heterochromatin-containing genes [33–36]. IBM2 contains a Bromo-Adjacent Homology (BAH) domain and an RNA recognition motif, and in the *Arabidopsis ibm2* mutant, genes containing heterochromatic introns show a transcription defect due to premature termination of transcripts at the heterochromatic regions. Intronic heterochromatin tends to repress expression of associated genes in both animals and plants [37–42]. However, in some circumstances establishment and maintenance of

heterochromatin within intronic regions are critical for transcriptional control of the associated genes required for environmental responses and development [28, 43, 44]. For example, in *A. thaliana*, maintenance of H3K9 methylation and DNA methylation of intronic TEs is important for transcription of the RPP7 gene, which confers resistance against a plant pathogen [45]. In winter wheat, vernalization induces DNA hypermethylation of the intron of *VRN-A1*, which promotes expression of the gene [43]. On the other hand, in oil palm, loss of DNA methylation of an intronic TE arising during tissue culture alters the splicing pattern of the associated gene, resulting in a developmental abnormality of the fruit [46]. These observations suggest that heterochromatin formation in intragenic, especially intronic TEs, may have functional relevance to transcriptional regulation of associated genes, and would also profoundly influence on gene diversification and evolution. Indeed, plant introns often encode regulatory elements for recruitment of transcription factors that alter chromatin states, which lead to both transcriptional repression and activation of developmental genes [47–50]. However, epigenetic states of introns at a genome-wide scale, their impacts on gene regulation, functions of genes bearing intronic heterochromatin, and their contribution to genetic diversity in plant genomes are not well understood.

The *Oryza sativa* genome is an ideal model for investigating interactions between genes and TEs, since 35% of the genome consists of TEs that are widely distributed in genic regions [51–53]. In this study, we investigated the genome-wide distribution of intronic heterochromatin and its impact on transcriptional control of genes in the rice genome. We found that over 10% of rice genes contain introns associated with repressive heterochromatin, which are involved in various biological processes. Transcription of genes bearing intronic heterochromatin as well as other genes without heterochromatic introns are affected by a loss of function of the conserved factor OsIBM2, which is essential for development and reproduction of rice. Rapid evolution of heterochromatic introns suggests their potential impacts on the evolution of gene sequences.

## Results

### Accumulation of heterochromatic introns in the rice genome

To investigate the epigenetic states of intronic regions in the rice genome (*Oryza sativa* L. ssp. *japonica* CV. Nipponbare), we performed whole-genome bisulfite sequence (WGBS) analysis using the mature rice leaf tissue. We specifically focused on detecting repressive heterochromatic states in intronic regions. Non-CG DNA methylation, especially CHG methylation, is well correlated with the heterochromatic state of histone modifications such as H3K9 methylation in plant genomes [20, 31, 54, 55]. Therefore, CHG methylation level was used as a proxy to define a heterochromatic state of chromatin within intronic regions (see methods for more details). We identified 5,809 introns within 4,227 gene models that contain heterochromatic domains in the rice genome (Fig 1A, S1 and S2 Tables). This is about 11% of gene models in rice genome (IRGSP-1.0; 37,866 gene models), and is 10-fold more abundant than in *Arabidopsis thaliana* (S1A Fig). Heterochromatic introns accumulate CG, CHG, and CHH methylation, as well as H3K9 di-methylation (H3K9me2; S1B Fig), similar to transposable elements (TEs; Fig 1B and 1C), indicating that they contain canonical heterochromatin. Loci containing heterochromatic introns are not biased toward repeat-rich pericentromeric regions, but are rather scattered throughout the rice chromosome arms (Fig 1D). RNA-seq analysis of the leaf tissue demonstrated that many of these loci are transcribed in the presence of intronic heterochromatin (Fig 1E), indicating that heterochromatic introns can co-exist within transcriptionally active genes in the rice genome.

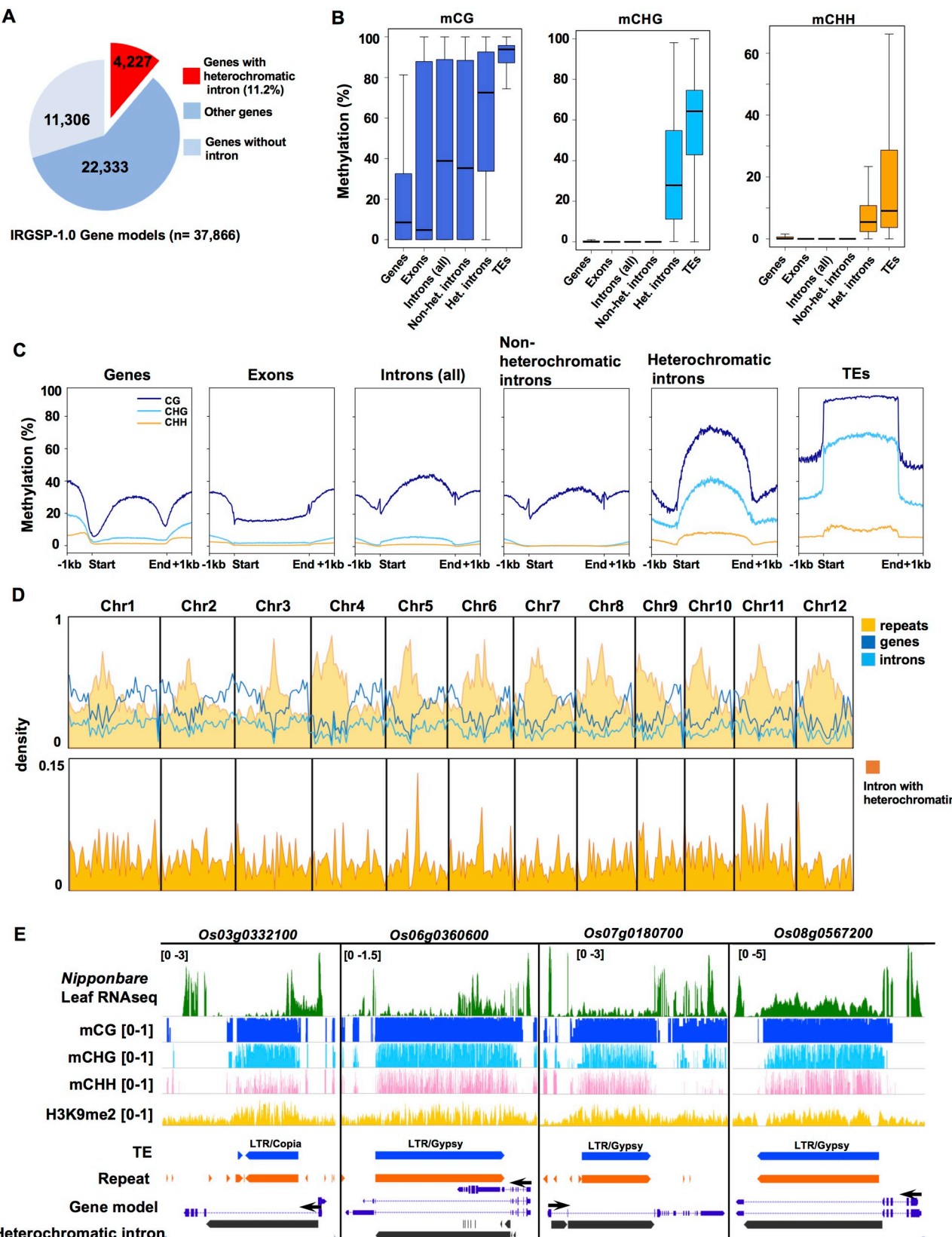

**Fig 1. Intronic heterochromatin in the rice genome.** (A) Rice IRGSP-1.0 gene models (n = 37,866) that contain heterochromatic domain in their intron (n = 4,227). (B) DNA methylation levels in CG (mCG), CHG (mCHG) and CHH (mCHH) contexts for indicated genome features. (C) Metaplots of DNA methylation in CG (blue), CHG (light blue) and CHH (orange) context for indicated genome features. (D) (Top) Density of repeats, genes, and intron sequences in 1MB bins in rice chromosomes. (Bottom) Density of introns with heterochromatic domains as above. (E) Representative rice genome loci containing heterochromatic domains within introns. Tracks: Top to bottom; RNAseq (Reads per Million are indicated at top left), mCG ratio (0 to1), mCHG ratio (0 to1), mCHH ratio (0 to1), H3K9me2 (RPM; 0 to 1), TE annotation (blue), repeats (orange), gene model (purple), introns containing heterochromatic domain (black). Black arrows indicate the orientation of coding sequence.

## Heterochromatin is enriched in promoter-proximal introns

In general, introns in the rice genome are longer than those in the *A. thaliana* genome (S2A Fig), which may be due to abundant repeat sequences in introns (13.9% of total intron sequence; S2B Fig). In particular, rice introns associated with CHG methylation tend to be longer (Fig 2A, S2C Fig) [34]. It has been reported that the first intron is generally longer than later introns in most of eukaryotic genomes, including that of rice [56] (Fig 2B). We found that heterochromatic introns are longer irrespective of their positions (Fig 2B). However, formation of heterochromatic introns is significantly biased toward the 5′-ends of rice genes ($p < 1.0e-6$ by a permutation test, Fig 2C), which is associated with accumulation of TEs in promoter-proximal introns (S2C Fig). This suggests that a preferential targeting of TEs toward the 5′-ends of rice genes might be a trigger for the formation of heterochromatin in promoter proximal introns.

## Many intronic TEs are short and hypomethylated in CHG context

Next, we investigated how the presence of TE affects heterochromatin formation within intronic regions. A set of manually curated TE annotations (n = 29,100, S3 Table) was analyzed for their locations in the genome. We found that about 7% (2,122/29,100) of TEs are located within intragenic regions (Fig 3A, S3A Fig), and that most of them (82%; 1,751/2,122) are present in introns (Fig 3B). TEs annotated as Miniature inverted-repeat transposable elements (MITE) are particularly enriched in intragenic regions (S3A Fig) consistent with previous studies [57, 58], while no strong orientation bias against the associated genes was observed in any of the TE families (S3B Fig). As expected, most of heterochromatic introns (84%; 4,886/5,809) are associated with TEs and other repeat sequences (Fig 3C). Interestingly, however, about 50% (980/1,967) of TE-containing introns do not overlap with the heterochromatic introns (Fig 3C), suggesting that intronic TEs are not always associated with heterochromatin. On the other hand, 16% (923/5,809) of heterochromatic introns are not associated with TEs or with repeat annotations, which is likely due to a spreading of heterochromatic modifications from neighboring chromatin (S3C Fig) [59, 60]. DNA methylation of intronic TEs seems to be maintained in the same manner as intergenic TEs, since methylation in intronic TEs is affected by mutations of maintenance methylase OsMET1, and the chromatin remodeler OsDDM1 (S3D Fig) [12, 21]. However, we found that a fraction of intronic TEs is hypomethylated especially in CHG, while distribution of CG, and CHH methylation levels among TEs is comparable between intergenic and intronic TEs, irrespective of the TE families (Fig 3D, S4 Fig). CHG-hypomethylated TEs are generally shorter than CHG-hypermethylated TEs (S5 Fig), and they are more abundant in introns (Fig 3E). A similar trend was observed in an analysis using a comprehensive MITE dataset [61] (153,751 TE sequences),which showed that short, CHG-hypomethylated MITEs (S6A and S6B Fig) are enriched in introns. Shorter TEs are likely degenerated or truncated TE sequences, on which relaxed epigenetic silencing may have resulted in a reduction of CHG methylation.

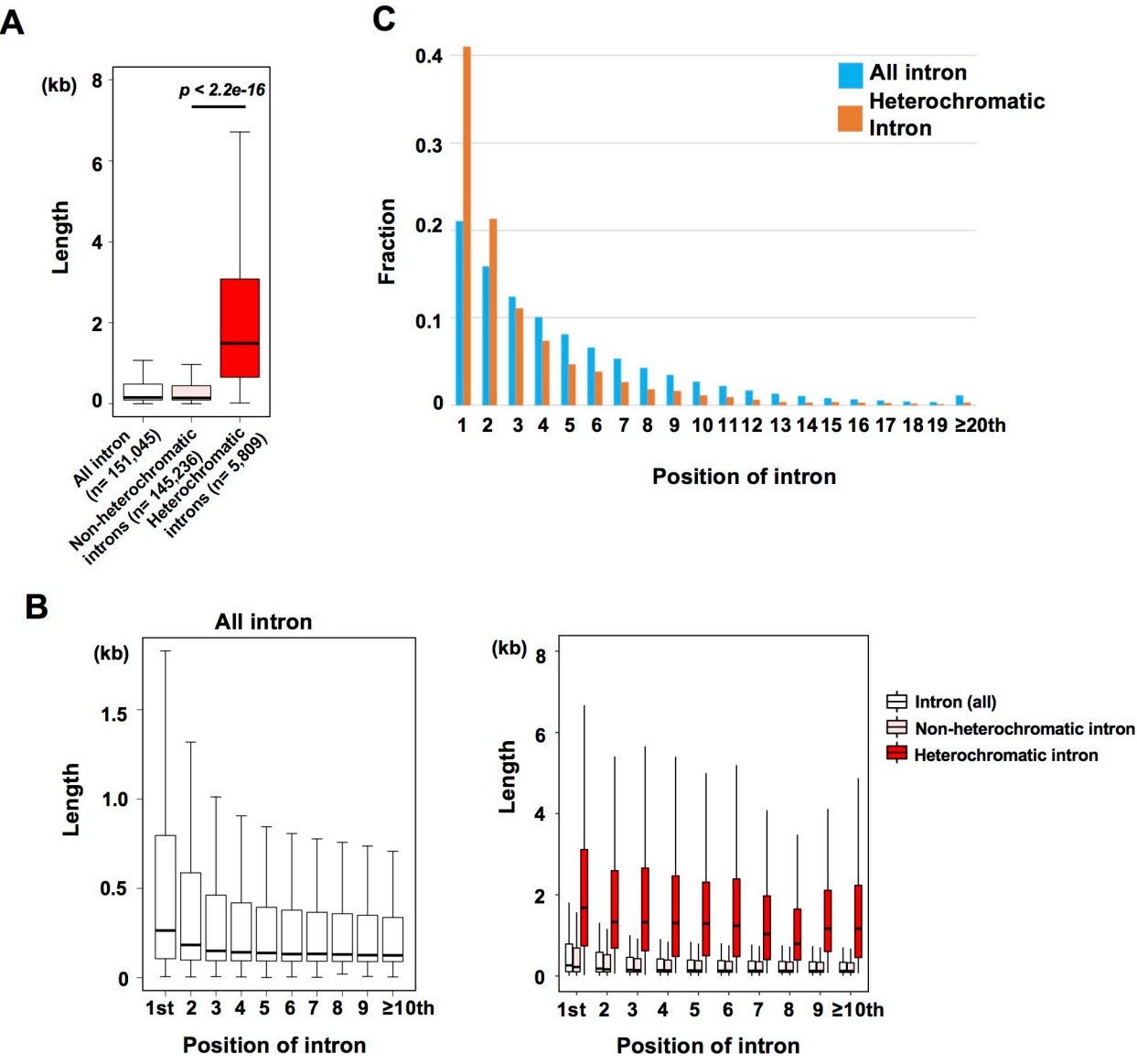

**Fig 2. Heterochromatin is enriched in promoter-proximal introns.** (A) Boxplots for length of normal and heterochromatic introns. Heterochromatic introns are significantly longer than introns without heterochromatic domains ($p$-value < 2.2e-16, Wilcoxon exact test). (B) (left) Intron position and length for all introns. (right) Intron position and length for all introns (white), non-heterochromatic introns (pink) and heterochromatic introns (red). (C) Enrichment of heterochromatin in promoter-proximal introns. Fraction of relative positions for all introns (n = 151,045), and heterochromatic introns (n = 6,086; the average position of heterochromatic introns was 3.02) are shown. Identical introns annotated in different positions in different splicing variants were independently counted.

## Heterochromatic introns are associated with genes involved in various biological functions

Rice genes with heterochromatic introns encode various proteins with enzymatic activities, including oxidoreductases and hydrolases, as well as with nucleotide-binding activities (S7A Fig). Gene Ontology (GO) enrichment analysis indicates that genes with heterochromatic introns are implicated in diverse functions, such as lipid/carbohydrate metabolic processes, post-embryonic and reproductive developmental processes, and cell death pathway, which is a manifestation of plant defense responses against pathogens (Fig 4A). On the other hand, GO

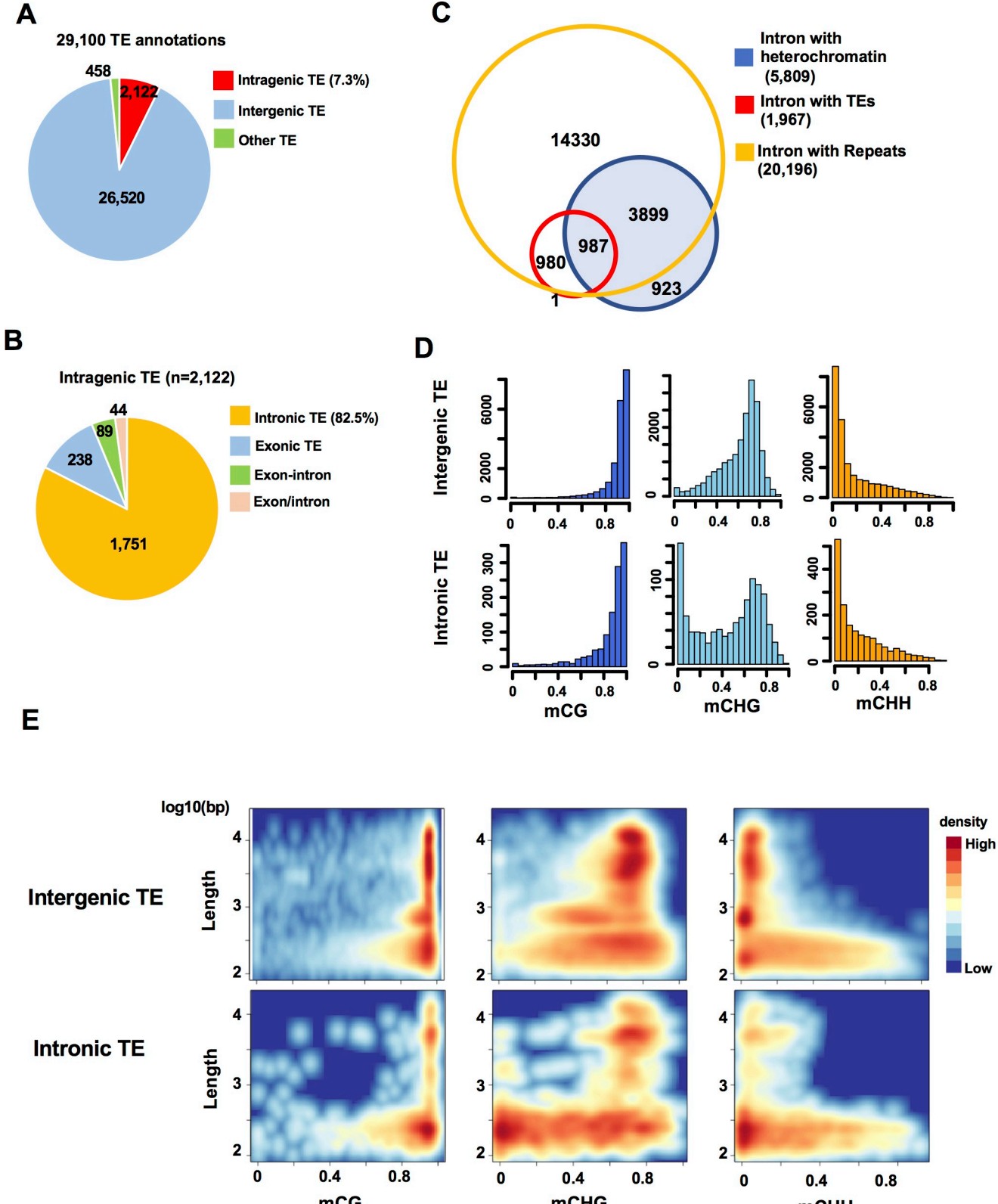

**Fig 3. Intronic TEs are short and hypomethylated.** (A) Classification of all TEs (n = 29,100) in the rice genome. "Other TEs" refers to TE annotations overlapping to both gene and intergenic regions. (B) Classification of intragenic TEs (n = 2,122) in the rice genome. "Exon-intron" refers to TE annotations

overlapping to both exon and intron. "Exon/intron" refers to TE annotations included in an exon of a gene/transcript model as well as in an intron of other gene/transcript models. (C) Venn diagram showing the number of overlapping introns containing heterochromatic domains (blue), TEs (red) and other repeats (yellow). (D) Histograms of the number of intergenic and intronic TEs and their methylation levels (0 to 1) in CG, CHG, and CHH contexts. TEs with methylation data at $\geq$ 5 Cs were included in the analysis. (E) Density plots showing length (log10) and methylation levels (0 to 1) of intergenic and intronic TEs in CG, CHG, and CHH contexts. TEs with methylation data at $\geq$ 5 Cs in each context were included in the analysis.

terms such as nitrogen biosynthetic/metabolic processes were depleted in the genes (S7B Fig). Our transcriptome analysis of mature rice leaf tissue showed that the expression levels of genes with heterochromatic introns were generally lower than those without heterochromatin (Fig 4B). We further examined expression patterns of rice genes in various developmental stages as well as in responses to environmental stimuli, using public microarray data in an expression atlas of rice genes and rice RNA-seq data [62, 63]. We calculated entropy values of gene expression patterns as a measure of specificity [64], which showed that genes with heterochromatic introns tend to have tissue-specific expression patterns, and are also responsive to plant hormones and environmental stresses (Fig 4C, S7C Fig). However, overall effect sizes of the values between gene with and without heterochromatic intron in the analyses were relatively small ($r < 0.1$), suggesting that expression profiles of genes with heterochromatic intron are not too different from genes without heterochromatin.

To understand the effects of heterochromatic introns on gene regulation in response to environmental signals, we searched for insertion/deletion polymorphisms in the intronic regions between Nipponbare (NB) and the indica-rice cultivar KASALATH (KAS) using whole-genome re-sequencing data [65]. In particular, we sought genes showing expression changes in response to JA, a plant hormone essential for development and also for both biotic and abiotic responses [66] (Fig 4C). Based on the genome re-sequencing data [67] and a public expression profile [63], we selected 12 JA-responsive loci (4 up-regulated, and 8 down-regulated loci after JA treatment; S8 and S9 Figs) that have large intronic deletions in the KASALATH genome corresponding to the regions showing heterochromatic state in the Nipponbare genome (S9 Fig). Consistent with the public expression profile, the JA-inducible genes *OsAOS2* [68, 69] as well as the 12 selected loci in NB showed expression changes in the root tissues upon JA treatment (S8 Fig). Several loci (4 out of 8 loci showing down-regulation in NB by JA treatment) in KAS showed reduced responses to JA treatment ($p>0.05$; $t$-test), whereas other loci including up-regulated genes showed essentially similar responses between NB and KAS, with variable degrees (S8 Fig). Thus, impacts of heterochromatic intron on the gene response and expression remain to be elucidated.

## A conserved epigenetic machinery regulates transcription of genes containing heterochromatin, and is essential for rice development

It has been shown in *A. thaliana* that transcription of genes with heterochromatic introns is regulated by a nuclear protein complex [33]. One of the proteins in the complex is Increased Bonsai Methylation 2 (IBM2), which contains a Bromo-Adjacent Homology (BAH) domain and an RNA recognition motif (S10 Fig)[34–36]. The rice homolog of IBM2 is encoded as a single-copy gene in the rice genome [34]. To examine whether it has a conserved function for transcription of genes with heterochromatic introns, we knocked down the transcript of the homologous gene (*Os01g0610300*; named as *OsIBM2*) using RNA interference (RNAi), targeting the 3′ end of the gene (Fig 5A). Among several independent T1 transformants, lines *#2* and *#16* showed a marked reduction of the transcript and were further investigated (Fig 5B). In addition, the CRISPR-Cas9 system was employed to obtain *OsIBM2* knock-out lines, which generated independent deletion mutant lines targeting either the BAH domain-encoding

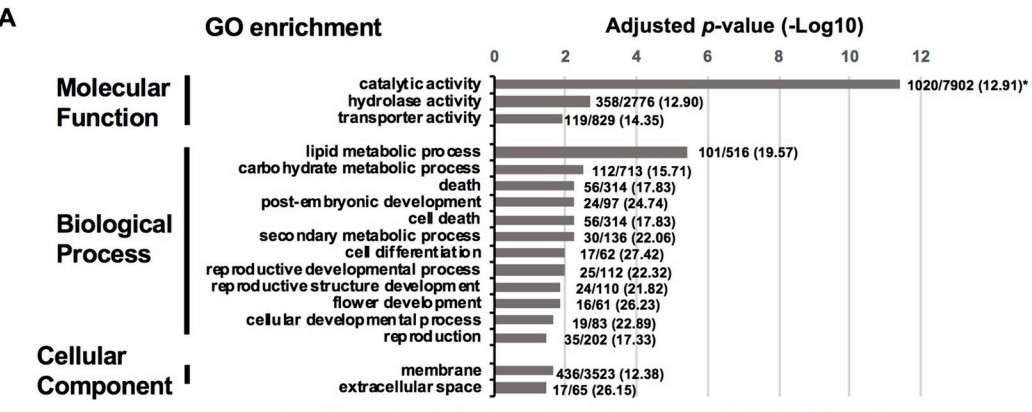

**A** GO enrichment

**B** Expression in leaf

**C** Development (48 stages)

Jasmonic Acid treatment (root)

ABA treatment (shoot)

Cold (root)

Flood (shoot)

**Fig 4. Genes with heterochromatic introns tend to show tissue-specific, and environment-responsive expression patterns.** (A) Gene Ontology enrichment for genes containing heterochromatic introns (2,449 genes out of 4,227 genes were analyzed for enrichment analysis. *p*-values were obtained by Fisher test with Hochberg adjustments (FDR < 0.05). GO terms (odds ratio; 95% Confidence Interval): catalytic activity (1.43; 1.31, 1.56), hydrolase activity (1.29; 1.14, 1.46), transporter activity (1.44; 1.17, 1.76), lipid metabolic process (2.10; 1.67, 2.63), carbohydrate metabolic process (1.60; 1.29, 1.97), death (1.85; 1.36, 2.49), post-embryonic development (2.79; 1.68, 4.50), cell death (1.85; 1.36, 2.49), secondary metabolic process (2.40; 1.55, 3.66), cell differentiation (3.21; 1.72, 5.73), reproductive developmental process (2.44; 1.50, 3.86), reproductive structure development (2.37; 1.44, 3.78), flower development (3.02; 1.59, 5.46), cellular developmental process (2.52; 1.42, 4.27), reproduction (1.78; 1.19, 2.59), membrane (1.23; 1.10, 1.38), extracellular space (3.01; 1.62, 5.34). (B) Expression levels of genes with heterochromatic introns in the leaf tissue measured by RNA-seq (Transcript per million; TPM > 0). *p*-values by Wilcoxon test are indicated. Effect size r = 0.046. (C) Tissue specificity of normal genes (pink) and heterochromatin-containing genes (red) in the rice developmental process, and specificities for Jasmonic Acid (JA) and abscisic acid (ABA) treatments, and stress treatments (Cold, Flood), measured by entropy values. *p*-values from the Wilcoxon test are indicated. Effect size (r) in each analysis: Development; 0.095, JA; 0.024, ABA; 0.031, Cold; 0.031, Flood, 0.037.

region (*g1#5* and *g1#27*) or a 3′ region downstream of the RRM encoding region (*g2#24*) (Fig 5C). Both RNAi and CRISPR-targeted mutants showed severe dwarfism and sterility (Fig 5D, S11A–S11C Fig). Particularly, mutants with deletions in the BAH domain (*g1#5* and *g1#27*) could not produce homozygous mutant seeds, suggesting an embryonic lethality of these alleles. Heterozygous mutants with a deletion in the 3′ region (*osibm2_g2#24*) could produce homozygous seeds, but the homozygous plants showed a complete sterility (S11C Fig), indicating that *OsIBM2* is essential for development and reproduction. Previous studies in *Arabidopsis ibm2* have shown that transcription at downstream of heterochromatic introns is reduced due to a premature termination of transcript within heterochromatic introns [34]. Therefore, we analyzed changes in accumulation of transcripts upstream and downstream of introns with both heterochromatic and non-heterochromatic state (total of 126,068 introns) in the rice genome. Our transcriptome analysis of the leaf tissues from both RNAi and CRISPR-Cas9 mutant lines detected 454 differentially expressed genes (DEGs) commonly in RNAi_#2, #16 and *osibm2_g#24* lines, which showed changes in transcripts downstream of introns compared with wild type (Fig 5E, S4 Table). Among DEGs, genes containing heterochromatic introns were significantly enriched (93 genes out of 454 DEGs (20.5%); *p* = 2.9e-17, Fisher's exact test). DEGs with heterochromatic introns showed a significant reduction of transcripts in the 3′ downstream of the heterochromatic intron (*p* = 1.0e-6, Tukey-Kramer test; S12B Fig), which was due to premature polyadenylation in the intronic regions (Fig 5F and 5G, S13, S14A–S14C Figs), similar to the phenotypes of the *Arabidopsis ibm2* [34, 70]. On the other hand, DEGs with normal introns showed less changes in their 3′ transcription (S12A and S12B Fig), suggesting that the mutation in *OsIBM2* results in transcription defects predominantly at heterochromatin-containing DEGs. We also searched for differentially expressed TEs in the *osibm2*. We detected only a few of them (23 TEs; 22 LTR, 1 DNA/En-Spm; 12 up-regulated, 11 down-regulated; S14D Fig), including 8 intronic TEs (3 TEs were associated with the DEGs containing heterochromatin; S14E Fig); some of these expression changes of TEs might be due to epigenetic changes during tissue culture transformation. The number of DEGs with and without heterochromatin that were detected by the RNA-seq analysis may have been underestimated, considering the partial loss of function of mutant alleles as well as the tissue-specific/environment-responsive expression profiles of heterochromatin-containing genes (Fig 4 and Fig 5). Indeed, additional RT-PCR analysis using RNAs from endosperm/embryo of *osibm2* showed that several heterochromatin-containing genes primarily expressed during reproductive development [71–76] were severely affected in *osibm2* (S11D Fig), even though they were not detected as DEGs in the RNA-seq of leaf tissues.

In *Arabidopsis*, the histone H3K9 demethylase gene, *IBM1*, contains heterochromatin in the 7th intron due to an insertion of organelle genome sequence, and *Arabidopsis ibm2* reduces expression of *IBM1*, which results in genome-wide accumulation of H3K9me2 and

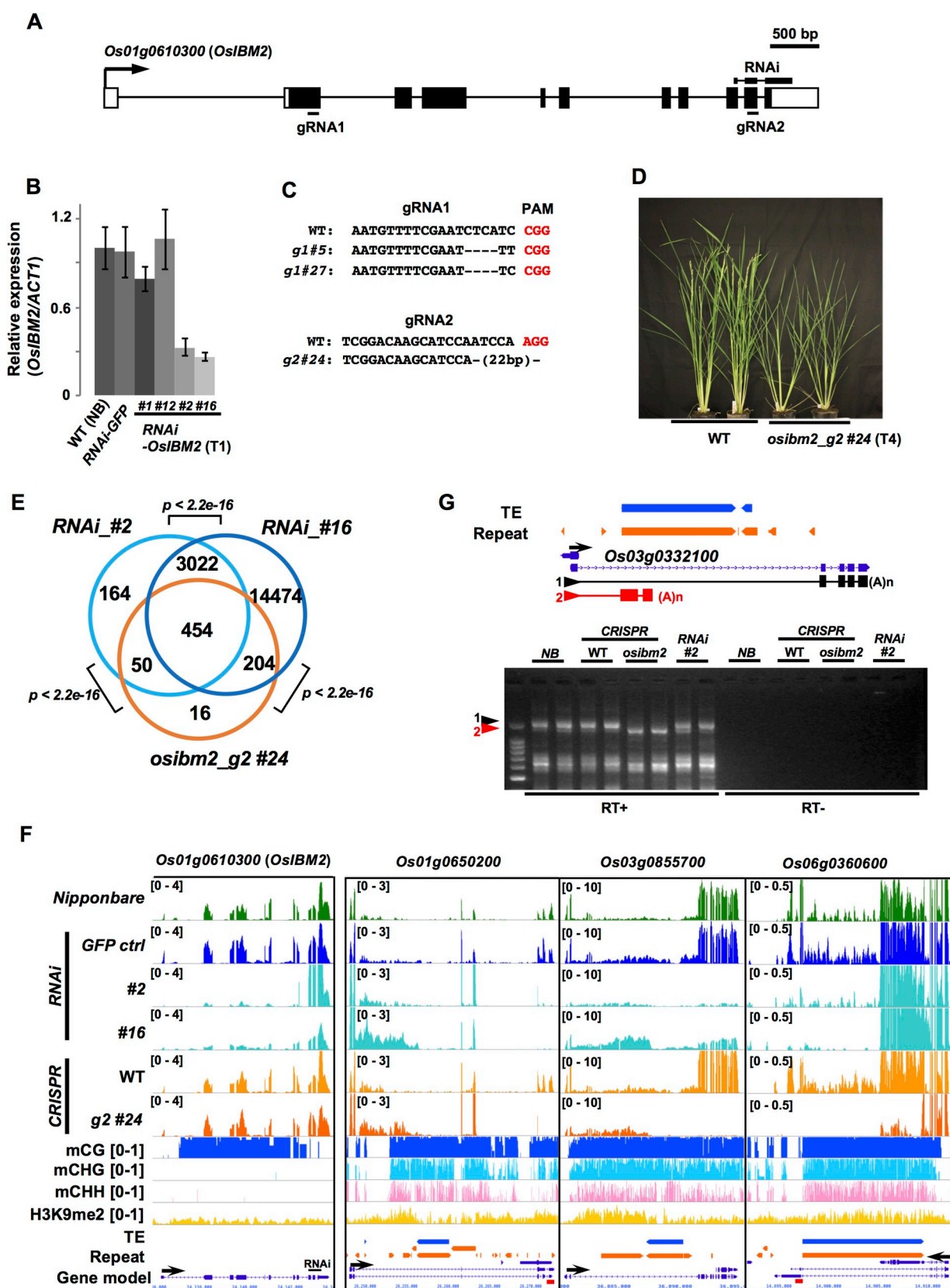

**Fig 5. OsIBM2 is required for rice development and transcription of heterochromatin-bearing genes.** (A) Gene structure of *Os01g0610300* (*OsIBM2*). Exons and untranslated regions are shown with black and white boxes, respectively. Regions designed for two gRNAs and hairpin RNA (RNAi) are also indicated. (B) qRT-PCR analysis of the expression of *OsIBM2* in 95-day-old leaf blade tissue of wild-type Nipponbare (NB), RNAi-*GFP* control line, and four RNAi-*IBM2* transgenic lines. Expression levels in each sample were normalized by *ACT1* expression levels, and the average of *OsIBM2/ACT1* in NB was set as 1. Bars represent means of three biological replicates ± S.E.M (*n* = 3). (C) Cas9 gRNAs and targeted deletions obtained in independent *osibm2* mutants. PAM: Protospacer Adjacent Motif. (D) Three-month-old rice plants of *osibm2_g2#24* and their segregating wild type siblings (WT; T4).(E) Venn diagrams of overlapping genes showing altered expression in RNAi #2, #16, and *osibm2_g2#24*. *P*-values for significance of overlaps were tested with Fisher's exact test.(F) Representative rice genome loci showing altered expression patterns in mutants of *OsIBM2*. Tracks: Top to bottom; RNAseq (Reads per Million are indicated in top left), mCG ratio (0 to1), mCHG ratio (0 to1), mCHH ratio (0 to1), H3K9me2 (RPM; 0 to 1), TE annotation (blue), repeats (orange), gene model (purple). The black arrow indicates the orientation of coding sequence. Red bars indicate primer positions for qPCR in S13B Fig. *OsIBM2* locus is shown as a validation of RNAi knock-down. (G) 3′ Rapid Amplification of cDNA Ends (RACE) of *Os03g0332100* containing intronic heterochromatin. Upper panel: Structure of *Os03g0332100* locus and polyadenylated mRNA variants detected by 3′ RACE. Exons and spliced introns confirmed by sequencing analysis are shown as black/red boxes and lines, respectively. Primer positions used for 3′ RACE are indicated by arrows. Lower panel: Gel picture of DNA fragments amplified by 3′ RACE. Two biological replicates for each genotype were examined. DNA fragments indicated by arrowheads were cloned and sequenced for at least 8 clones, and the representative sequences supported with more than 3 clones are shown in the upper panel. NB: Nipponbare; *osibm2*: *osibm2_g2#24*; WT: wild type segregants of *osibm2*; (A)n: polyadenylation.

non-CG methylation at genic regions [77, 78]. We therefore scrutinized, by WGBS analysis of *osibm2*, whether *OsIBM2* regulates non-CG methylation in the rice genome using the CRISPR mutant *osibm2_g#24*. We found that DNA methylation patterns in CG and non-CG contexts were nearly identical in genic as well as intergenic regions between *osibm2* and WT (S15 Fig). In the rice genome, two *IBM1* homologs, *OsJMJ718* (MSU ID: *Os09g22540*; RAP ID: *Os09g03 93200*) and *OsJMJ719* (MSU ID: *Os02g01940*; RAP ID: *Os02g0109400*, *Os02g0109501*), have been identified (S16A and S16B Fig) [79]. We found that one of the *IBM1* homologs, *OsJMJ718* contains heterochromatin in the last intron (S16A Fig), although it was not identified as a commonly affected gene among *OsIBM2* mutants (significant transcript changes were detected in RNAi_#2 and #16; *q* < 0.01). The less significant effects of the *OsIBM2* mutation on *OsJMJ718* expression and genome-wide non-CG methylation may be due to a partial loss of function of *OsIBM2* in the mutants, or to functional redundancy of *OsJMJ718* and *OsJMJ719*. Alternatively, the *OsJMJ18* transcript may be more resistant to the effects of heterochromatin that is downstream of the jmjC domain-coding sequence, compared with the *A*.

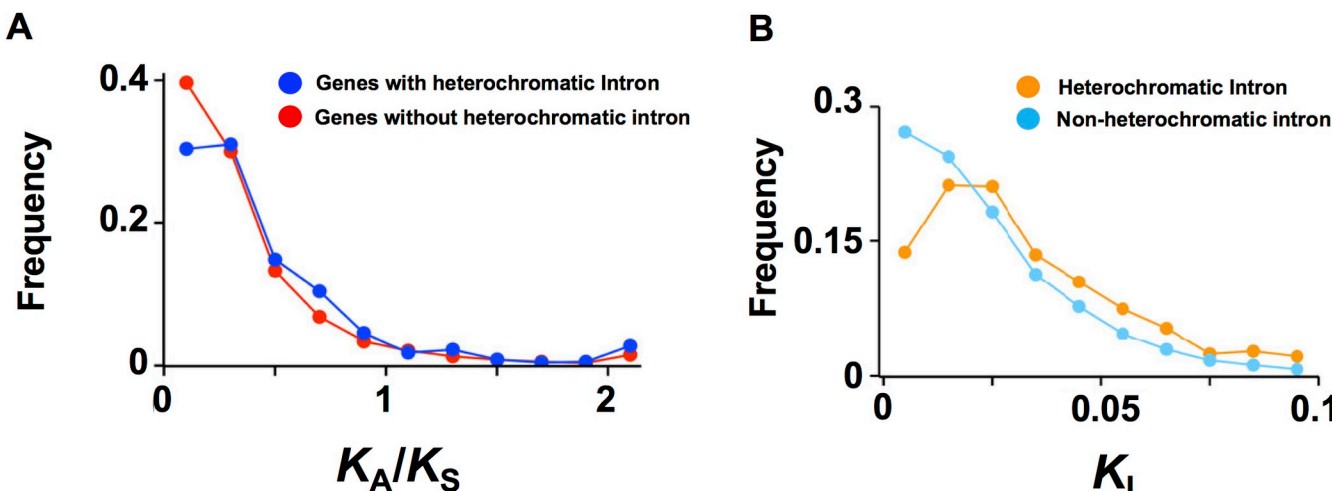

**Fig 6. Patterns of nucleotide substitution rates between *Oryza sativa* and *Oryza meridionalis*.** (A) Frequency distributions of $K_A/K_S$ values. Blue and red plots represent genes with and without heterochromatic introns, respectively. (B) Frequency distributions of $K_I$ values. Orange and light blue plots represent heterochromatic introns and heterochromatin-free introns, respectively.

*thaliana IBM1*, which has the heterochromatic intron in the middle of jmjC domain-coding sequence (S16C Fig).

## Rapid evolution of heterochromatic intron

To understand how heterochromatin formation affects gene evolution in the rice genome, we further investigated the pattern of nucleotide substitutions in rice genes with heterochromatic introns. We first tested whether the degrees of selective constraints are similar between genes with and without heterochromatic introns in *O. sativa*. To this end, we compared the genome sequence of *O. sativa* with that of a close wild relative, *O. meridionalis* [80]. We predicted the orthologs in *O. meridionalis* and calculated the rate of nucleotide substitutions (Materials and Methods). Although our previous study of the *A. thaliana* genome did not find a significant difference [28], we found a relaxation of selective constraints in genes with heterochromatic introns in the rice genome (Fig 6A), where the ratio of nonsynonymous substitution rates to synonymous substitution rates ($K_A/K_S$) was 0.473 ($n = 928$), compared to 0.384 ($n = 10,456$) in genes without heterochromatic introns ($P < 10^{-5}$ by a permutation test). This indicates that heterochromatic introns would be deleterious for genes under high levels of selective constraint.

We further investigated the pattern of nucleotide substitutions in introns ($K_I$). Even though we excluded repeat sequences for the inter-species comparison, $K_I$ values of heterochromatic introns showed higher base substitution rates (0.0325; $n = 627$) than non-heterochromatic introns (0.0242; $n = 35,354$; $P < 10^{-5}$ by a permutation test) (Fig 6B). This indicates that heterochromatic introns have evolved more rapidly than heterochromatin-free introns, suggesting an acceleration of intronic sequence divergence associated with heterochromatin formation.

## Discussion

In this study, we revealed the genome-wide distribution of heterochromatic introns in the rice genome, which contains heterochromatic introns in approximately 11% of the genes. The underlying molecular mechanisms that allow the presence of repressive heterochromatin within actively transcribed regions are still unclear [37]. However, our study demonstrated that the conserved epigenetic factor OsIBM2 is critical for production of proper mRNA through heterochromatic introns in dozens of loci in the rice genome (Fig 5, S11D Fig). In addition, many genes without heterochromatic introns are also affected by *OsIBM2* mutation in the leaf tissue (Fig 5), suggesting the profound impact of the loss of function of OsIBM2. Rice mutants of major epigenetic regulators, including OsMET1, OsCMT3, OsDRM and OsDDM1, have been shown to exhibit severe developmental defects such as embryonic/seedling lethality and sterility [14, 20, 81–83], while phenotypes of mutants of these genes in *A. thaliana* are relatively mild and plants are essentially viable [11, 84–87]. The difference likely stems from the genome structure of rice, where abundant TEs are distributed along gene-rich chromosome arms [51]. The close association of TEs with genes would make genes more susceptible to epigenetic changes in nearby TEs. In *A. thaliana*, *ibm2* plants are still fertile [34], while rice *osibm2* results in severe developmental defects and sterility (Fig 5, S11 Fig). This suggests that in the rice genome, in addition to the maintenance of heterochromatic states by DNA methyltransferases, transcriptional regulation of genes by OsIBM2 affects rice development and reproduction. For plant genomes harboring abundant intragenic heterochromatin, gene regulation mechanisms involving IBM2 would be more vital [29–31].

Insertion of TEs in intronic regions often results in repression of associated genes due to accumulation of repressive epigenetic marks. In rice, insertions of MITE in an intron cause

repression of the Elongated Uppermost Internode (EUI) gene, which is due to siRNA production from the intronic TEs [88]. In *Arabidopsis* and *Capsella* natural strains, insertions of TEs in the intron of flowering repressor gene *Flowering Locus C* (*FLC*) downregulate its expression, and induce early-flowering phenotypes [41, 59, 89]. Consistent with these reports, we found that rice genes with heterochromatic introns tend to show lower expression in the leaf tissue (Fig 4B). Alternatively, genes expressed at lower levels may tolerate those insertions [90]. On the other hand, our analysis of JA-responsive loci with insertion/deletion polymorphisms in heterochromatic introns suggested that responsiveness of genes to the hormone are largely unaffected by the presence/absence of heterochromatin in introns (S8 and S9 Figs). However, further comprehensive analyses are required to fully understand the impacts of intronic heterochromatin on gene regulation during environmental responses.

Longer first introns are a universal feature of eukaryotic gene structure [56]. The first intron sequence is more conserved than the later introns in animal genomes [91, 92]. In plants, enhancement of gene expression by intronic sequences, known as intron-mediated enhancement (IME), is associated with specific sequence motifs enriched in the first intron [93]. In rice, the first introns are required for the higher expression of tubulin genes [94, 95]. Intriguingly, intronic heterochromatin is significantly enriched in first and second introns which are associated with the accumulation of TEs in these introns (Fig 2, S2C Fig). Many TEs are known to target the 5′ end of genes [7, 96, 97], while insertions into the exons in the 5′ ends of genes would be selected against, which may result in the accumulation of TEs in promoter-proximal introns. Insertion of TEs and formation of repressive chromatin may physically disrupt or override transcription enhancer functions of the promoter-proximal introns, which may contribute to lower expression of the associated genes (Fig 4B). Additionally, the inserted TEs may provide novel regulatory sequences such as transcription factor binding sites [6, 22], allowing genes to acquire tissue-specific, or environment-responsive expression properties (Fig 4). The degree of selective constraints and tissue specificity are negatively correlated in *Arabidopsis* species [98]. Consistent with this, we observed that genes with heterochromatic introns tend to be expressed in a tissue-specific manner, and to show a lower degree of selective constraints than the other genes (Fig 6). We also observed that heterochromatic intron sequences show higher evolution rates (Fig 6), likely due to the higher mutation rates of methylated cytosine residues [99]. Thus, the formation of heterochromatin in intronic regions may contribute to the divergence of gene sequences.

The association of repetitive elements with genes is most prominent in disease-resistance gene (R gene) loci in plant genomes, which would accelerate gene diversification by enhancing recombination, and by shuffling and duplication of the sequences [6]. Indeed, R-genes are significantly overrepresented in genes with heterochromatic introns (119 out of 689 R-genes; 2.8% of 4,227 genes with heterochromatic introns; *p* = 1.66e-6, Fisher's exact test). Also, our GO analysis showed that heterochromatic introns are enriched in genes involved in the cell death pathway, which is provoked during plant immune responses mediated by R-genes [100, 101]. Acquiring repressive chromatin by TE insertions within intronic regions may also contribute to reduced expression of R genes, which may be advantageous for the prevention of autoimmune responses in the absence of pathogens [102].

A recent study showed that wild rice genomes tend to accumulate TEs in genic regions, while cultivated rice genomes show depletion of TEs from genic regions including introns [103]. This has likely occurred independently in the genomes of several cultivars [103]. This convergent loss of genic TE sequences in cultivar genomes may be a result of selective pressure against long heterochromatic TEs in the genic regions during domestication and selection (Fig 3). Alternatively, under uniform growing conditions in a nutrition-rich environment, inbreeding cultivar genomes may have gradually lost environment-responsive regulatory elements

associated with genic TEs. In contrast, longer introns with TE insertions in wild rice genomes may be adaptive for dynamic transcription changes in the fluctuating natural environment. Indeed, recent studies in budding yeast demonstrated that the presence of introns promotes survival under starvation conditions, while the introns are dispensable in a nutrient-rich environment [104, 105]. Intron sequences in plant genomes may have more profound impacts on genome evolution and plant adaptation than previously thought.

## Methods

### Rice genome annotations

Annotations of *Oryza sativa* genome, version IRGSP v1.0, locus/transcript/repeat annotations (IRGSP-1.0_representative_2015-03-31_2) were retrieved from RAP-DB (http://rapdb.dna. affrc.go.jp/) [106]. We identified TEs in the *Japonica* rice genome using RepeatMasker (ver. 4.0.5; http://www.repeatmasker.org). Repbase library (ver. 20140131) [107] was downloaded and used as a repeat library. We ran RepeatMasker with the default parameters and screened putative TE segments. We first excluded non-TE repeats such as simple repeats, rRNAs and satellite DNAs. We then further filtered out the following results; 1) the hit regions covering <70% of the total length of the repeats in the library, 2) the length of the hit regions is < 100 bp, 3) nucleotide divergence between the hit region and the repeat in the library is >20%. The list of TEs is in S3 Table. MITE annotation was retrieved from the P-MITE database [61], and used for a BLASTN [108] search of the IRGSP genome with a cutoff e-value of 1e-40. MITE sequences with identical lengths to query sequences having no mismatch and no gap (153,751 sequences) were used for further analysis. Chip-seq data for H3K9me2 was obtained from [109]. BS-seq data for *osmet1* and *osddm1* were obtained from [12, 21], respectively. Rice seed core collections (World Rice Core Collection; WRC) were obtained from Genebank Project, National Agriculture and Food Research Organization (NARO; https://www.gene.affrc.go.jp/ databases-core_collections_wr.php).

### Rice transgenic lines

All rice plants used in this study were grown in growth chambers under short-day condition (10 hours light/ 14 hours dark cycles) at 30˚C during daytime and 25˚C during the night. For RNAi knock-down of the *OsIBM2* mRNA, about 500 bp of the cDNA sequence of *OsIBM2* was cloned into pANDA vector [110]. A partial GFP sequence was used as a control RNAi vector. Wild-type Nipponbare calli were transformed with the RNAi vector at InPlanta Innovations (Yokohama, Japan) or at our laboratory, and more than 15 independent T1 transformants for each vector were obtained. For CRISPR-Cas9 knock-out of *OsIBM2*, two guide RNAs (S5 Table) were designed and cloned into pHUE411 (Addgene #62203) by GoldenGate Mix (NEB), and transformed into rice calli with a standard agrobacterium transformation method. Gene targeting events were detected by digestion with HpaII (gRNA1) or HaeIII (gRNA2), and were confirmed by Sanger sequencing. For *osibm2_g2#24*, the absence of the pHUE411 vector and fixation of the mutation (Fig 5C) were confirmed at T3. Segregating wild type (WT) and homozygous T4 plants were used for further analyses.

All oligonucleotides used in this study are listed in S5 Table.

### Bisulfite sequencing and data analysis

For Whole Genome Bisulfite-Sequencing (WGBS) analyses, we used genomic DNA of Nipponbare, *osibm2_g2#24* (T4), and wild-type segregants of *osibm2_g2#24* (T4) isolated from three-month-old mature leaf tissues with Nucleon PhytoPure (GE). An Illumina Sequencing

libraries (125 bp paired-end for Nipponbare, 150 bp paired-end for *osibm2_g2#24* and Wild-type) were constructed using the PBAT method [111] and sequenced at OIST Sequencing Center (SQC). Raw reads were trimmed by Trimmomatic [112] with parameters; HEAD-CROP:10 SLIDINGWINDOW:4:20 MINLEN:50. Remaining paired reads were mapped to rice genome IRGSP v1.0 with Bismark (v0.19.0) [113] with parameters; -N 1—pbat -ambiguo-us -R 10 -un—score_min L,0,-0.6. Unmapped reads together with dropped single-end reads from trimming were further mapped to the rice genome as single-end reads with parameters; -N 1—pbat -ambiguous -R 10—score_min L,0,-0.6, for R1, and -N 1 -ambiguous -R 10 -un—score_min L,0,-0.6, for R2. Methylation reports from paired and single reads were merged with bedtools [114]. Only uniquely mapped reads were used for further analysis, and C bases covered by fewer than 3 reads, and also Cs more than 100 (Cs with unnaturally high coverage; top ~0.01% of covered Cs) were excluded. Methylcytosines were identified by binomial test [115], with the bisulfite conversion rate estimated by mapping sequencing reads to the rice chloroplast genome. Methylation levels were calculated using the ratio of $#C/(#C + #T)$ as described in [116]. Methylcytosine domain containing consecutive $\geq 5$ mCHG with $\geq 0.5$ methylation on average was considered as heterochromatic domain. Boxplots, sequence density, and metaplots for DNA methylation were generated with deeptools [117], Microsoft Excel, and R. A summary of WGBS analyses is shown in S6 Table.

## GO analysis

GO analysis for enrichment was performed using the AgriGO website [118] and significant terms were extracted by Fisher's exact test with Hochberg adjustment (FDR<0.05). GO term depletion analysis was performed with TopGO (https://rdrr.io/bioc/topGO/) using Fisher's exact test. Protein classes were determined using the Panther database [119].

## Expression data analysis

Micro-array data and RNA-seq data were retrieved from RiceXpro database [62] and TENOR [63]. For gene expression of developmental stages, gene expression profiles of 48 rice developmental stages/tissues were used for calculation of entropy value of each gene. For gene expression profiles of stress/hormone treatment conditions, gene expression data at following time points were used for calculation of entropy value of each gene: Jasmonic Acid, ABA, Cold, Drought treatments; 0, 1, 3, 6, 12, 24 hours, Flood treatment; 0, 1, 3, 6, 12, 24, 72 hours, Osmotic stress; 0, 1, 3, 6, 12 hours, High/Low phosphate treatments; 0, 1, 5, 10 days. Entropy (modified H) was calculated with ROKU function in TCC package in R [120].

## Genome sequencing data analysis for indel identification

Genome resequencing data of KASALATH genomes was retrieved from [65] and mapped to IRGSP-1.0 using bowtie2-2.2.2 [121]. Candidate loci for intronic deletion in the KASAKLATH genome were searched based on the INDEL data retrieved from [122]. The presence of deletion was confirmed by PCR.

## Jasmonic Acid treatment and gene expression analysis

The rice strains, Nipponbare and KASALATH (WRC 2), were germinated on plates. After 6 days, seedlings were transferred to 15 mL plastic tubes and grown hydroponically in 1/10 Murashige-Skoog (MS) media for 4 days in a growth chamber as described above. Plants were transferred to 1/10 MS media containing the final 100 μM Jasmonic Acid (JA, SIGMA) and 0.02% DMSO, or 0.02% DMSO as a mock treatment. After 6 hours of treatments, total RNA

was extracted from the roots using Maxwell 16 LEV Plat RNA kit (Promega), and Quantitative RT-PCR (qRT-PCR) was performed for analysis of gene expression.

## RNA-seq analysis

For RNA-seq analysis, total RNA from the leaf tissues was isolated with Maxwell 16 LEV Plat RNA kit (Promega). Two biological replicates for Nipponbare (NB), GFP-RNAi control lines (T2), RNAi *#2* lines (T2), *osibm2_g2#24* lines (T4), and wild-type segregants of *osibm2_g2#24* (WT; T4) were prepared. An additional NB line was used for comparison with the single RNAi *#16* (T2) line. Illumina RNA-Seq libraries (150bp paired-end) were prepared and sequenced at the OIST Sequencing Center. Raw reads were trimmed with Trimmomatic with the following parameters; HEADCROP:10 LEADING:15 TRAILING:15 SLIDINGWINDOW:10:15 MIN-LEN:25. Remaining paired reads were mapped to rice genome IRGSP v1.0 with Hisat2 [123] with parameters;—min-intronlen 20—max-intronlen 20000. Exon and splicing junction information was specified by the annotation retrieved from RAP-DB to prepare a genome index for Hisat2. A summary of RNA-seq analysis is shown in S7 Table. Reads mapped to rDNA and tRNA were removed with bedtools. For visualization of RNA-seq read tracks, read duplication was removed with samtools [124], and Reads per million (RPM) for 1 bp bin was calculated with deeptools. The read tracks were visualized in Integrated Genome Browser [125]. For estimation of expression level, reads mapped on transcript annotations were counted with the featureCounts function in Rsubread [126] with parameters; allowMultiOverlap = TRUE, minOverlap = 1, fracOverlap = 0, countMultiMappingReads = FALSE, and used for Transcript Per Million (TPM) calculations for each gene model (Fig 4B). Expression changes in RNAi (RNAi_*#2* and _*#16*) and CRISPR knock-out lines (*osibm2_g2#24*) were analyzed based on the methods in [34]. Transcripts mapped to pre- and post- introns (n = 126,068) in each gene model were counted by featureCounts. Ratio of read counts (mapped reads in pre-intron: mapped reads in post-intron) of two biological replicates of each genotype (RNAi_*#2* lines vs RNAi_GFP lines, *osibm2-g2#24* lines (T4) vs wild-type segregants of *osibm2-g2#24* (WT; T4)) were tested to detect changes in the expression pattern, by employing logistic regression analysis with *p*-value correction by Benjamini-Hochberg (BH) method for multiple testing. Changes in gene expression between *RNAi_#16* (one replicate) and control NB were detected by binominal test with *p*-value correction as above. Data sets with $q \leq 0.01$ were considered as significantly changed in downstream transcription (both up- and down-regulated loci in 3′ region). A relative 5′/3′ ratio of transcripts mapped to up- and down-stream of introns was calculated as described previously [34]. Differential expression analysis of TEs was performed by DESeq2 [127] using mapped read data by Hisat2 (*osibm2-g2#24* lines (T4) vs wild-type segregants of *osibm2-g2#24* (WT; T4)).

Quantitative RT-PCR (qRT-PCR) and 3′ RACE were performed as described in [34].

## Nucleotide substitution analysis

To reveal patterns of nucleotide substitutions in genes with heterochromatic introns, we compared nucleotide sequences of *O. sativa* and *O. meridionalis* [80]. Putative orthologs were identified using GenomeThreader [128] with mRNAs of *O. sativa* to find orthologs in *O. meridionalis*, with the following parameters; *-minmatchlen 18 -seedlength 16 -exdrop 2*. When multiple orthologs were detected for an mRNA, it was discarded. If no ortholog was detected, we incremented the parameter *-exdrop* by one. This process was repeated until a single ortholog was detected or until the parameter *-exdrop* was less than or equal to 5. We further screened orthologs in which the exon-intron structures were conserved between the orthologs in 80% of their nucleotide sequences after alignments with CLUSTALW2 [129]. Nonsynonymous and synonymous

nucleotide substitution rates ($K_A$ and $K_S$, respectively) were calculated using the Nei and Gojo-bori method [130]. We discarded genes with $K_S > 0.1$. We also calculated nucleotide substitution rates in introns as *p*-distance.

## Supporting information

**S1 Fig. Heterochromatic introns in *Arabidopsis thaliana* and rice genomes.** (A) *Arabidopsis thaliana* genes (TAIR10) containing intron with heterochromatic domains. (B) Heatmap showing accumulation of H3K9 di-methylation on genome features in the rice genome. Data from [109] were used for the analysis.
(PDF)

**S2 Fig. Length of introns in *Arabidopsis thaliana* and rice genomes.** (A) A comparison of intron length between *Arabidopsis thaliana* (n = 127,836; average 169.0 bp) and *Oryza sativa* (n = 126,068; average 446.9 bp). (B) Fraction of repetitive elements in intronic regions of the rice genome. (C) Enrichment of heterochromatin and TEs in promoter-proximal introns. Fractions of all intron (n = 151,045), and heterochromatic introns (n = 6,086), and TE-containing introns (n = 1,982) are shown in the relative positions. Identical intronic regions annotated in different positions in different splicing variants were independently counted.
(PDF)

**S3 Fig. TE families in rice introns.** (A) Fraction of TE families in the intronic regions of the *Oryza sativa* genome. (B) Orientation of intronic TE insertion against gene annotations in each TE family. No significant orientation bias was observed in the TE families ($p > 0.01$; two-sided binominal test). (C) Metaplots of DNA methylation in CG, CHG and CHH contexts for heterochromatic introns with TEs and repeats (n = 4,886), heterochromatic introns without repeat (n = 923), and non-heterochromatic introns (n = 145,235). (D) Heatmap of methylation profiles of intronic TEs in wild-type *O. sativa* and mutants of *OsMET1* (*met1*) and of *OsDDM1* (*ddm1*) at CG, CHG, and CHH-contexts.
(PDF)

**S4 Fig. DNA methylation of rice intergenic and intronic TEs.** Histograms of the number of representative intergenic and intronic TE families (>20 copies in each category) and their methylation levels (0 to 1) in CG, CHG, and CHH contexts. TEs with methylation data at $\geq 5$ Cs were analyzed.
(PDF)

**S5 Fig. Length and DNA methylation of intronic TEs.** Boxplots showing length of representative intergenic and intronic TE families (>10 copies in each category) and their methylation levels in CG (high; mCG $\geq 0.9$, low; mCG $< 0.9$), CHG (high; mCHG $\geq 0.2$, low; mCHG $< 0.2$), and CHH (high; mCHH $\geq 0.1$, low; mCHH $< 0.1$). * $p < 0.05$, ** $p < 0.01$, *** $p < 0.001$, Wilcoxon exact test. N.S.: no significance, $p \geq 0.05$. TEs with methylation data at $\geq 5$ Cs were analyzed.
(PDF)

**S6 Fig. DNA methylation of MITEs in rice introns.** (A) Histograms of the number of representative intergenic and intronic MITEs (data retrieved from the P-MITE database [61] and their methylation levels (0 to 1) in CG, CHG, and CHH contexts. TEs with methylation data at $\geq 5$ Cs were used in the analysis. (B) Density plots showing length (log10) and methylation levels (0 to 1) of intergenic and intronic MITEs in CG, CHG, and CHH contexts.
(PDF)

**S7 Fig. Protein classes and expression changes of genes containing heterochromatic introns.** (A) Protein classes defined by the Panther database [119]. 1,407 of 4,227 genes containing heterochromatic introns matching the database are indicated. (B) Gene Ontology depletion for genes containing heterochromatic introns. *P*-values were obtained by Fisher test, and terms with FDR < 0.05 are indicated. (C) Expression changes of all genes and genes with or without heterochromatic introns by various stress treatments. Specificity of the responses to given treatments were measured as entropy values. *P*-values from Wilcoxon exact test are indicated. Effect size (r) in each analysis: Low phosphate; 0.024, High phosphate; 0.020, Drought; 0.007, Osmotic stress; 0.009.
(PDF)

**S8 Fig. JA response of genes in Nipponbare (NB) and KASALATH (KAS) with structural variations in heterochromatic intron.** (A) Heatmap showing expression levels of the indicated genes after Jasmonic Acid (JA) treatment in the Nipponbare root. Expression data were obtained from TENOR [63]. (B) Quantitative RT-PCR (qRT-PCR) analysis of genes before (pre-treatment), and after JA (JA treatment). *OsAOS2* was included as a control for JA-dependent induction of expression. Relative expression levels in each sample were normalized by *UBQ1* expression levels, and the average of expression values in pre-treatment NB samples was set as 1, and plotted as dots (*n* = 6) with blue (NB) and yellow (KAS). The large dots and bars represent means of 6 biological replicates ± standard deviation (S. D.). *P*-values were obtained by *t*-test.
(PDF)

**S9 Fig. Structural variations of heterochromatic introns in Nipponbare and KASALATH strains.** Insertion/deletion polymorphisms in Nipponbare and KASALATH. Tracks: Top to bottom: mCG ratio (0 to1), mCHG ratio (0 to1), mCHH ratio (0 to1), genome-resequencing data coverage (0 to 30) [65], repeats (orange), TE annotation (blue), gene model (purple). Structural variations detected by PCR are indicated under the tracks as gel pictures. Red arrows indicate the primer positions used for PCR amplifications shown in the gel panel. The region used for qRT-PCR is indicated as red bar.
(PDF)

**S10 Fig. Amino acid alignment of homologs of OsIBM2.** Amino acid alignment of homologs of OsIBM2 in plants based on [34]. Bromo-Adjacent Homology (BAH) domain and RNA-Recognition Motif (RRM) are framed with a blue line. Arrows indicate regions designed for guide RNAs used for CRISPR-Cas9 mediated deletion. At; *Arabidopsis thaliana*: Zm; *Zea mays*: Os; *Oryza sativa*: Sb; *Sorghum bicolor*: Pt; *Populus trichocarpa*: Rc; *Ricinus communis*.
(PDF)

**S11 Fig. Developmental phenotypes of *osibm2* mutants.** (A) Whole plant picture of three-month-old Nipponbare (left), RNAi_*#2* line (middle) and RNAi_GFP control line (right). (B) Close-up pictures of seeds set in Nipponbare and RNAi lines (T1). (C) A close-up picture of seeds set in *osibm2_g2#24* and their segregating wild-type siblings (WT; T4). White bar: 1 cm. (D) RT-PCR analysis of gene expression in endosperm and embryo of Nipponbare and *osibm2*. RNAs from ~10 DAF (Days After Fertilization) developing endosperm and embryo of *osibm2_g2 #24* (T2) were used for the analysis.
(PDF)

**S12 Fig. Expression changes in genes containing heterochromatic introns in *osibm2*.** (A) (Top) DNA methylation levels of differentially expressed genes (DEGs) with heterochromatic introns (n = 93), DEGs without heterochromatic intron (n = 361), and non DEGs (n = 20293)

in Nipponbare background. (middle) DNA methylation difference in *osibm2* (*osibm2_ g2 #24*) and wild type at loci as above. (Bottom) H3K9 methylation levels at loci as above. (B) 5′/3′ ratio of transcripts mapped to up- and down-stream of introns relative to wild type. RNA-seq data from *osibm2_ g2 #24* and WT (wild-type segregants of *osibm2*) were used. In each locus, the 5′/3′ ratio of a representative transcript variant with TPM >1 was used for calculation. Bars represent the means of DEGs with heterochromatic introns (n = 68), DEGs without heterochromatic intron (n = 335), and randomly selected 300 nonDEG loci ± S.E.M. *p*-values were obtained by Tukey-Kramer test.
(PDF)

**S13 Fig. Expression changes of genes in *osibm2*.** (A) Representative rice genome loci showing altered expression patterns in mutants of *OsIBM2*. Tracks; Top to bottom: RNAseq (Reads per Million are indicated at top left), mCG ratio (0 to1), mCHG ratio (0 to1), mCHH ratio (0 to1), H3K9me2 (RPM; 0 to 1), TE annotation (blue), repeats (orange), gene model (purple). The black arrow indicates the orientation of coding sequence. (B) Quantitative RT-PCR (qRT-PCR) analysis of expression of genes containing heterochromatic introns in *osibm2_g2#24* (*osibm2*) and WT (wild-type segregants of *osibm2*). Primer positions are indicated in Fig 5F and S13A Fig as red bars. Expression levels in each sample were normalized by *UBQ1* expression levels, and the average of *OsIBM2/UBQ1* in WT was set as 1. Bars represent the means of three biological replicates ± S. D. (*n* = 3).
(PDF)

**S14 Fig. 3′ Rapid Amplification of cDNA Ends (RACE) of genes containing heterochromatin in mutants of *OsIBM2*.** (A) 3′ RACE of *Os01g0650200*. Upper panel: Structure of *Os01g0650200* locus and polyadenylated mRNA variants detected by 3′ RACE. Exons and spliced introns confirmed by sequencing analysis are shown as black/red boxes and lines, respectively. Primer positions used for 3′ RACE are indicated by arrowheads. Lower panel: Gel picture of DNA fragments amplified by 3′ RACE. Two biological replicates for each genotype were examined. DNA fragments indicated by arrowheads were cloned and sequenced at least for 8 clones, and the representative sequences supported with more than 3 clones are shown in the upper panel. The black arrow indicates the orientation of coding sequence. NB: Nipponbare; *osibm2*: *osibm2_g2#24*; WT: wild-type segregants of *osibm2*; (A)n: polyadenylation. (B) 3′ RACE of *Os06g0360600* as in (A). (C) 3′ RACE of *Os08g0567200* as in (A). (D) The number of TEs showing expression changes in *osibm2_g2#24* (*osibm2*). 22 LTR TEs, and 1DNA/En-Spm showed significant changes (*q*<0.05) of both up-regulation (12 TEs) and down-regulation (11 TEs). (E) Rice genome loci showing altered expression patterns of intronic TEs in mutants of *OsIBM2*. Tracks; Top to bottom: RNAseq (Reads per Million are indicated at top left), mCG ratio (0 to1), mCHG ratio (0 to1), mCHH ratio (0 to1), H3K9me2 (RPM; 0 to 1), TE annotation (blue), repeats (orange), gene model (purple). The black arrow indicates the orientation of coding sequence.
(PDF)

**S15 Fig. DNA methylation in *osibm2*.** (A) Genome-wide DNA methylation in *osibm2_g2#24* (*osibm2*, T4) and their wild type segregating siblings (WT, T4) in CG, CHG and CHH contexts for each chromosome. Average methylation levels in 1 MB bins were plotted. (B) Metaplots of DNA methylation in *osibm2_g2#24* (*osibm2*) and their wild-type segregating siblings (WT) in CG, CHG and CHH contexts for indicated genome features.
(PDF)

**S16 Fig. Rice homologs of the *Arabidopsis* H3K9 demethylase *IBM1*.** Genome loci for *OsJMJ718* (*Os09g0393200*) (A) and *OsJMJ719* (*Os02g0109400, Os02G0109501*) (B). RNA-seq,

DNA methylation and H3K9me2 tracks are shown as in S13 Fig. (C) An alignment of amino acids sequences of *A. thaliana* IBM1 (At_IBM1) and OsJMJ718. The amino acid sequence of the N-terminal part of OsJMJ718 is predicted based on RNA-seq reads in this study. The alignment was generated by CLUSTAL W [131]. Jumonji-C (JmjC) domains predicted by SMART [132] are circled with blue lines. Positions of heterochromatic introns are indicated by red arrowheads.
(PDF)

**S1 Table. Genes containing heterochromatic introns.**
(XLSX)

**S2 Table. Chromosomal positions of heterochromatic introns.**
(XLSX)

**S3 Table. Transposon annotation used in this study.**
(XLSX)

**S4 Table. Genes showing expression changes in *osibm2* mutants.**
(XLSX)

**S5 Table. Primers used in the study.**
(XLSX)

**S6 Table. A summary table for Whole Genome Bisulfite Sequencing (WGBS) analysis.**
(XLSX)

**S7 Table. A summary table for RNA-seq analysis.**
(XLSX)

**S1 Data. Numerical data used to generate Figures.**
(XLSX)

## Acknowledgments

We thank the Genebank project, NARO, for rice seed collection. We thank OIST SQC for BS-seq and RNA-seq analysis, and Drs. Yoshiki Habu, and Reina Komiya for critical reading of the manuscript. We also thank Dr. Steven D. Aird for editing the manuscript.

## Author Contributions

**Conceptualization:** Nino A. Espinas, Hidetoshi Saze.

**Data curation:** Le Ngoc Tu, Shohei Takuno, Hidetoshi Saze.

**Formal analysis:** Nino A. Espinas, Le Ngoc Tu, Leonardo Furci, Yasuka Shimajiri, Yoshiko Harukawa, Saori Miura, Shohei Takuno.

**Funding acquisition:** Hidetoshi Saze.

**Investigation:** Nino A. Espinas, Le Ngoc Tu, Leonardo Furci, Yasuka Shimajiri, Yoshiko Harukawa, Saori Miura, Shohei Takuno.

**Project administration:** Hidetoshi Saze.

**Supervision:** Hidetoshi Saze.

**Visualization:** Le Ngoc Tu, Leonardo Furci, Shohei Takuno, Hidetoshi Saze.

**Writing – original draft:** Nino A. Espinas, Leonardo Furci, Shohei Takuno, Hidetoshi Saze.

**Writing – review & editing:** Nino A. Espinas, Le Ngoc Tu, Shohei Takuno, Hidetoshi Saze.

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
