## [Decision Letter · Decision Letter 0]

18 Jul 2019

Dear Dr Saze,

Thank you very much for submitting your Research Article entitled 'Genome-wide distribution of intronic heterochromatin impacts gene transcription and sequence divergence in the rice genome' to PLOS Genetics. Your manuscript was fully evaluated at the editorial level and by independent peer reviewers. The reviewers appreciated the attention to an important problem, but raised some substantial concerns about the current manuscript. Based on the reviews, we will not be able to accept this version of the manuscript, but we would be willing to review again a much-revised version. We cannot, of course, promise publication at that time.

As you will see from the detailed comments of the reviewers, they all consider the work as potentially interesting, but in the current state, they did not find it suitable for publishing. They have a number of suggestions for shifting the focus within the paper, for specifying or adding details, and for considering alternative interpretations. Specifically, they raise quite some points about the statistical handling of the data, and addressing these issues might require adjustment of some of the statements correspondingly. In addition, they suggest some improvements for the data presentation. We hope that these comments are helpful to revise the manuscript.

If you decide to revise the manuscript for further consideration at PLOS Genetics, please aim to resubmit within the next 60 days, unless it will take extra time to address the concerns of the reviewers, in which case we would appreciate an expected resubmission date by email to plosgenetics@plos.org.

[LINK]

We are sorry that we cannot be more positive about your manuscript at this stage. Please do not hesitate to contact us if you have any concerns or questions.

Yours sincerely,

Ortrun Mittelsten Scheid

Associate Editor

PLOS Genetics

Wendy Bickmore

Section Editor: Epigenetics

PLOS Genetics

Reviewer's Responses to Questions

**Comments to the Authors:**

Reviewer #1: The authors characterise heterochromatic introns (defined as introns with 5 or more consecutive mCHG with average methylation >=0.5) in the rice genome, and demonstrate that the OsIBM2 gene plays a similar role to its Arabidopsis counterpart in the transcription through heterochromatic regions, but unlike in Arabidopsis OsIBM2 mutations have severe phenotypic consequences. The manuscript is clear and interesting. My comments are therefore mostly technical. The conclusions about regulatory roles of intronic heterochromatin can be de-emphasised, while the data from OsIBM2 knock-downs and knock-out is very nice and could have a greater place in the abstract.

Major comments:

1. The evidence for the regulatory roles of intronic heterochromatin is too thin to include it as a conclusion in the abstract. The sentence in the summary (“may have regulatory roles...”) is more accurate, and should be preferred.

2. Part of the evidence comes from a GO analysis. GO enrichment results are in general difficult to interpret, but it is even more difficult here as the fold-enrichments and number of genes of each category are not reported. The broad term “catalytic activity” is significantly enriched, but how many of the 2,500 genes with heterochromatic introns are concerned?

3. Additional evidence comes from the entropy of gene expression. Although there are significant differences between genes with and without heterochromatic introns, the magnitudes of the difference seem very small. The text should reflect this.

4. The evidence around the JA responsiveness is very weak. From just 4 cultivars and 3 genes with no consistent response patterns across cultivars, I believe the carefully-worded conclusion that “these results suggest that structural variations of heretochromatic introns may have impacts on gene responses to environmental signals” is still too strong: it may also have no impact. One would need a lot more cultivars to start deriving trends, and/or a lot more JA-responsive genes. I recommend keeping the results in the manuscripts but noting that they do not provide evidence. The authors rightly note that non-intronic sequence polymorphims may also explain the altered JA response; trans-factors may also be at play.

In presenting this data, plotting individual points rather than mean + sd would be preferable. The use of asterisks is better kept for statistical significance than category of structural variation. Please also give the results of the multiple-testing-corrected statistical tests.

Minor comments:

1. Please provide a summary table for the sequencing (WGBS and RNA-seq): number of reads sequenced and mapped, bisulfite conversion rate, median cytosine coverage, etc. It will help in quickly assessing study design and robustness of the underlying data.

2. The mapping strategy for the bisulfite data may cause a small amount of double-counting: if the reads of a pair overlap each other, and map to a SNP (or PCR error), they would not be aligned in paired end (no mismatches allowed) but would both be successfully aligned in the second round as single-end reads with 1 mismatch. The cytosines in the overlapping region would then be counted twice. This should be rare and is unlikely to affect the results, but I’d recommend relaxing the mapping parameters in the paired-end round. Here again the summary table of mapping efficiency/... would help in evaluating the mapping strategy.

3. Cytosines with low coverage (0-2) are excluded, but it may also be worth to exclude outliers at the upper end: regions of unnaturally high coverage may be chloroplast insertions in the genome assembly or, particularly relevant to this study, repeated elements that are not well represented in the assembly.

4. It is unusual to deduplicate RNA-seq reads, in the absence of UMIs it risks introducing biases (for a discussion see https://sequencing.qcfail.com/articles/libraries-can-contain-technical-duplication/ and https://www.biostars.org/p/55648/). Did the libraries require this? If they did, any results obtained after deduplication should be taken with a pinch of salt.

5. It is also unusual to collapse biological RNA-seq replicates (l. 524-528). Please keep the replication in the analysis even if it means relaxing the significance threshold to look at an OK number of genes.

6. TPM usually refers to Transcripts Per Million rather than Tags Per Million. If the authors mean tags per million, the cpm (counts per million) unit would be clearer.

7. Plots in Fig 4B and C are the showing the same thing, only with a different axis-scale. It’s a good idea to plot log values of cpms/tpms rather than non-logged counts, as in Fig4C. If the style of Fig4B is desired, a violin plot would advantageously summarise all the information into one graph.

boxplots

8. In the context of their study, the authors discuss the documented link between R genes and TEs. Could this be specifically addressed? For instance, are R genes over-represented in the set of genes with intronic-heterochromatin?

Reviewer #2: This manuscript provides a comprehensive analysis of introns with heterochromatin in rice. Given that we know very little about this in larger genomes, and that the authors did comparative analysis with other rice cultivars, and that they included mutant analysis, I was initially excited about these results. The idea that intronic TE insertions may be an important source regulator information is indeed intriguing, but I do not think the authors have made the case, nor have to considered numerous alternative interpretations of their data. The mutant analysis was particularly disappointing, as only a tiny proportion of any of the genes’ expression was affected in the expected way and a more comprehensive analysis of the RNAseq of the mutants was missing. In the end, the authors ended up with anecdotes that may or may not support what is clearly a favored hypothesis. However, particularly when it comes to TEs and epigenetics, we must always consider the null hypothesis, which is that most TEs most of the time do not provide a selective advantage to their host. The authors are have made some intriguing arguments based on their data, but, in the end, it is not convincing.

Line 68: “often acquire regulatory functions for surrounding genes”. Unclear what this means.

Line 113: “introns at genome-wide” should read “introns at a genome-wide”

Line 159: or preferential targeting? Many TEs appear to preferentially target the 5’ ends of genes.

Line 165: Hopefully, this data set will be made available.

Line 178: This is very likely the case. Are these repetitive or single copy? Do they form hairpins? Are they helitrons, which are notoriously difficult to identify?

Line 246: I’m not sure I find these data particularly convincing. It is a small number of genes and, as the authors state, there certainly could be other causal sequence polymorphisms. It would have been more convincing with more examples, perhaps of other enriched terms? Also, the error bars from the RT-PCR are pretty large.

Line 279: I’m having a hard time interpreting this. “detected 198 genes both with and without heterochromatic introns that showed changes in transcripts downstream of introns (27%; 54/198 genes with heterochromatic intron, expression of which commonly changed in RNAi_#2, #16 and osibm2_g#24 lines; Table S3)”. So a total of 198 showed changes and some were with and some were without heterochromatic introns. And only 27% of the total were heterochromatic. Is this statistically significant? Since 11% of introns are heterochromatic, I guess it could be, but the argument should be made here. And the percent of heterochromatic introns that are affected was only 54/4,150, or 1.3%. I may be missing something, but this result does not appear to represent a trend.

Line 283: But the same could be for the non-heterochromatin intron genes.

Line 288: Why not do the control, which are genes whose downstream expression went down even though they didn’t have any heterochromatin?

Line 289: I disagree, what the data suggests is that a very small fraction of genes with and without heterochromatin are affected when OsIBM2 is knocked down.

Line 295: Which allele was this?

Line 326: In plants with large genomes, many gene models do not actually correspond to genes. Rather, they represent Pack-MULEs and helitrons, that capture gene fragments. If they were inadvertently counted, then this would explain an overall relaxation in selection. Alternatively, genes under relaxed selective contraints may simply be able to tolerate these insertions.

Line 332: Don't you mean exon here? You can’t get Ka/Ks from introns.

Line 343: I’m not convinced of this. The mutants could cause lethality for a variety reasons, and only a minority of genes that were changed downstream of the introns had heterochronic introns, as the authors point out. Further, I’m guessing that expression of a very large number of genes besides these were affected in the mutant.

Line 359: Alternatively, some genes, such as those expressed at lower levels or under certain conditions may simply tolerate these insertions.

Line 376: There is also ample evidence that many TEs target the 5’ ends of genes, and insertions into the exons in the 5’ ends of genes would be selected against.

Line 389: I don’t think so. It just shows that neutral sequences evolve rapidly, and methylation makes them evolve more rapidly.

Reviewer #3: The manuscript by Espinas and colleagues reports on the description of heterochromatic introns (HIs) in the rice genome. The authors have performed WGBS to reveal introns with heterochromatic signature (DNA methylation in all sequence contexts) and found around 4000 genes harboring such HIs. They further analyzed the epigenetic control of these introns, and obtained mutants (RNAI lines and CRISPR mutant) for IBM2. The mutants are sterile suggesting embryonic lethality. However only 200 common genes are affected in the 3 lines analyzed. While the study is of interest I think the manuscript would benefit from a more complete description of some results.

1. From Figure 3C it seems that repeats are enriched (compared to TEs) at HIs. Could the authors precise how were these repeats annotated? Concerning the authors' own annotation of TEs, how does it compare with published ones? (What % of the genome is covered?).

2. Figure 4: what are the functional categories depleted for HIs?

3. Study of polymorphimss at HIs: why did the authors limit their analysis to 3 varietes when 3,000 rice genomes are available? Did they compare with the 12 rice species with assembled genomes?

4. The functional evidence based on only one gene (line 245) should be interpreted with great caution. Only CRISPR excision of the HI or introgression line could properly address this question. This validation would be beyond the scope of this study.

5. IBM2 mutant analysis is not described in the abstract, not in the introduction. It should be mentioned as it represents an important part of the manuscript. Did the authors investigate the putative target genes involved in the sterile phenotype? What about TE expression in this mutant? I think this part of the work will be of interest to many rice researchers and could be described in more details. Fore instance how to explain the strong phenotype given that osibm2 has no impact on DNA methylation? Could the difference in overexpressed genes in the 3 lines be stochastic and depend on the level of IBM2 remaining expression (line 16 clearly the most affected)?

Minor comments:

Line 105 reference to be edited.

Line 106: please specify that this epiallele (Karma) is revealed under in vitro culture conditions

0s11g0229300 could be shown in Figure 1.

**Have all data underlying the figures and results presented in the manuscript been provided?**

Reviewer #1: Yes

Reviewer #2: Yes

Reviewer #3: Yes

PLOS authors have the option to publish the peer review history of their article (what does this mean?). If published, this will include your full peer review and any attached files.

Reviewer #1: Yes: Quentin Gouil

Reviewer #2: No

Reviewer #3: Yes: Marie Mirouze

---

## [Decision Letter · Decision Letter 1]

3 Jan 2020

Dear Dr Saze,

Thank you very much for submitting the revision of your Research Article entitled 'Transcriptional regulation of genes bearing intronic heterochromatin in the rice genome' to PLOS Genetics. Two of the previous reviewers accepted to re-review the revision, and both agree that the manuscript has substantially improved. The main concern has been addressed, but there are some aspects that still need minor revisions. As both reviewers state, providing the TE annotation data in an accessible format is essential. The entropy analysis in connection with the GO terms should be better specified, and some specific questions need to be addressed, as you will see from the detailed comments. There are also several suggestions for text edits.

We therefore ask you to modify the manuscript according to the review recommendations before we can consider your manuscript for acceptance. Your revisions should address the specific points made by each reviewer.

[LINK]

Yours sincerely,

Ortrun Mittelsten Scheid

Associate Editor

PLOS Genetics

Wendy Bickmore

Section Editor: Epigenetics

PLOS Genetics

Reviewer's Responses to Questions

**Comments to the Authors:**

Reviewer #1: The authors have made a substantial effort to address the comments. The data are better presented and the toned-down conclusions are more accurate.

I would still like to see a mention of effect size, and not only statistical significance, in the description of the GO enrichment and entropy analyses. For example, an 18% enrichment in cell death terms seems quite small. Overall it seems that the GO terms of heretochromatic intron-containing genes are not too different from the rest of the genes. For the entropy (and overall expression), the changes also appear to be modest. For the R gene enrichment, please specify the “background” R gene percentage.

As requested by reviewer 2, the TE annotation should be deposited rather than relying on direct requests.

There is a small formatting error on the p-value on l.161.

Reviewer #2: Overall, the authors have done a good job addressing the concerns I had. However, I do have some remaining issues that should be addressed before publication.

General note: Since the locations of the TEs is essential for replicating these experiments, I do not think that making the TEs “available on request” is sufficient. The data should be provided as a supplemental data set in an easily convertible format (not a PDF). Also, please be sure to address my confusion concerning entropy values for the various conditions.

Line 39. Not sure what “basal functions” means here.

Line 53. And yet most genes with h-introns are not affected by the mutation.

Line 66. “ species with small genome” should be small genomes

Line 95. You should say what IBM2 is here.

Line 144. should read, “similar to”

Line 210. Figs. should be Fig.

Line 223. I find this a bit confusing. How is the entropy for the treatments work? Were multiple tissues for each treatment examined? If not, how was entropy calculated? By just comparing treated and untreated? If you are talking about “responsiveness” wouldn’t you want to look at changes in expression (up or down?).

Line 229. looked for, or sought, rather than sought for.

Line 275. So expression of the downstream exons relative to upstream exons, or relative to w.t.?

Line 276. Given that the degree of knockdown for the two RNAi lines are nearly identical, why is there a massive difference between the two with respect to differential expression of downstream exons? Also, just to be clear, these are all DEG with respect to the ratio of 5’ to 3’ exons, not all the DEGs?

Line 281. Since I can’t tell where the primers are and the “truncated” transcript looks like a smear, I am not convinced that figure 5G shows premature polyadenylation. Figure 5F is confusing, since it shows that 610300 (OsIBM) has a loss of 5’ exons in plants targeting the 3’ exons. However, the overall trend, for downstream exons to be lost (presumably through premature polyadenylation) is clear and convincing, and the examples in Figure S14 are more convincing (perhaps one of these should be used in place of the one in the main figure).

Line 287. I’m assuming that none of these were associated with premature polyadenylation?

Line 316. Indeed, the TE insertion is even missing from the A. lyrate homolog of IBM1, so the global effects of IBM2 are almost certainly contingent on a random insertion into IBM1 in A. thaliana.

Line 333. should read constraint. Interestingly, the same appears to be true for upstream insertions of TEs in maize.

Line 341. What if you masked the methylated regions?

Line 352. This is hardly surprising given that many genes are presumably indirectly affected by the mutant.

Line 363. Not to quibble, but I often find this kind of argument frustrating. The implication is that heterochromatin formed at TEs is “functional” because premature polyadenylation at the TE in the IBM2 mutant occurs. However, I would argue that the IBM2 protein simply masks the presence of the TE so that it can be tolerated. What would convince me otherwise would evidence that the TE insertion in wild type plants has an effect on any aspect of the phenotype.

Line 383. I never bought that argument, unless you think that maize has a far more sophisticated regulatory apparatus than does arabidopsis.

Line 420. Selective pressure against?

Line 425. Yes, but aren’t these wild rices at least partly outcrossing?

Line 520. How could you get entropy values for this if you only looked at one tissue?

**Have all data underlying the figures and results presented in the manuscript been provided?**

Reviewer #1: No: The TE annotation, central to the manuscript, should be deposited as pointed out by reviewer 2. "Upon request" is not sufficient, as outlined by the Plos Genetics guidelines

Reviewer #2: No: They need to provide the TE data as a supplemental, not "on request".

PLOS authors have the option to publish the peer review history of their article (what does this mean?). If published, this will include your full peer review and any attached files.

Reviewer #1: Yes: Quentin Gouil

Reviewer #2: No

---

## [Editor Report · Decision Letter 2]

28 Jan 2020

Dear Dr Saze,

We were glad to read that the reviewers' comments were helpful to improve the manuscript. Thank you for the careful revision and considering the input. We are pleased to inform you that your manuscript entitled "Transcriptional regulation of genes bearing intronic heterochromatin in the rice genome" has been editorially accepted for publication in PLOS Genetics. Congratulations!

Yours sincerely,

Ortrun Mittelsten Scheid

Associate Editor

PLOS Genetics

Wendy Bickmore

Section Editor: Epigenetics

PLOS Genetics

Comments from the reviewers (if applicable):

**Data Deposition**

http://datadryad.org/submit?journalID=pgenetics&manu=PGENETICS-D-19-00869R2

**Press Queries**

---

## [Editor Report · Acceptance letter]

11 Mar 2020

PGENETICS-D-19-00869R2 

Transcriptional regulation of genes bearing intronic heterochromatin in the rice genome 

Dear Dr Saze, 

We are pleased to inform you that your manuscript entitled "Transcriptional regulation of genes bearing intronic heterochromatin in the rice genome" has been formally accepted for publication in PLOS Genetics! Your manuscript is now with our production department and you will be notified of the publication date in due course.

With kind regards,

Kaitlin Butler

PLOS Genetics

On behalf of:
